# Circulating Biomarkers of CDK4/6 Inhibitors Response in Hormone Receptor Positive and HER2 Negative Breast Cancer

**DOI:** 10.3390/cancers13112640

**Published:** 2021-05-27

**Authors:** Ilenia Migliaccio, Angela Leo, Francesca Galardi, Cristina Guarducci, Giulio Maria Fusco, Matteo Benelli, Angelo Di Leo, Laura Biganzoli, Luca Malorni

**Affiliations:** 1“Sandro Pitigliani” Translational Research Unit, Hospital of Prato, Azienda USL Toscana Centro, 59100 Prato, Italy; angela.leo@uslcentro.toscana.it (A.L.); francesca.galardi@uslcentro.toscana.it (F.G.); giuliomaria.fusco@uslcentro.toscana.it (G.M.F.); luca.malorni@uslcentro.toscana.it (L.M.); 2Department of Medical Oncology, Dana-Farber Cancer Institute, Harvard Medical School, Boston, MA 02215, USA; cristina_guarducci@dfci.harvard.edu; 3Bioinformatics Unit, Hospital of Prato, Azienda USL Toscana Centro, 59100 Prato, Italy; matteo.benelli@uslcentro.toscana.it; 4“Sandro Pitigliani” Department of Medical Oncology, Hospital of Prato, Azienda USL Toscana Centro, 59100 Prato, Italy; angelo.dileo@uslcentro.toscana.it (A.D.L.); laura.biganzoli@uslcentro.toscana.it (L.B.)

**Keywords:** CDK4/6 inhibitors, circulating biomarkers, liquid biopsy, therapy resistance, breast cancer

## Abstract

**Simple Summary:**

Biomarkers found in the blood of patients with hormone receptor positive and HER2 negative metastatic breast cancer are being investigated to understand how patients respond to treatments. Circulating biomarkers have the potential advantage of giving important information with a simple withdrawal of peripheral blood. Here, we review and discuss the recent achievements in the development of circulating biomarkers in patients with metastatic breast cancer treated with CDK4/6 inhibitors and endocrine therapy.

**Abstract:**

CDK4/6 inhibitors (CDK4/6i) and endocrine therapy are the standard treatment for patients with hormone receptor-positive and HER2 negative (HR+/HER2−) metastatic breast cancer. Patients might show intrinsic and acquired resistance, which leads to treatment failure and progression. Circulating biomarkers have the potential advantages of recognizing patients who might not respond to treatment, monitoring treatment effects and identifying markers of acquired resistance during tumor progression with a simple withdrawal of peripheral blood. Genomic alterations on circulating tumor DNA and serum thymidine kinase activity, but also circulating tumor cells, epigenetic or exosome markers are currently being tested as markers of CDK4/6i treatment response, even though none of these have been integrated into clinical practice. In this review, we discuss the recent advancements in the development of circulating biomarkers of CDK4/6i response in patients with HR+/HER2−breast cancer.

## 1. Introduction

The majority of breast cancers (BC) (around 70%) expresses hormone receptors (HR), either estrogen (ER) or progesterone (PR) receptors or both and is responsive to endocrine therapies (ET), including aromatase inhibitors (AIs), selective estrogen receptor modulators (SERMs) and selective estrogen receptor degraders (SERDs) like fulvestrant [1]. Women with early stage ER positive BC have a 20-year risk of distant recurrence ranging from 22% to 52%, depending on the nodal status [2]. Currently, inhibitors of cyclin-dependent kinases 4 and 6 (CDK4/6i), namely abemaciclib, palbociclib and ribociclib, administered with ET, represent the standard for the treatment of patients with HR positive and HER2 negative (HR+/HER2−) metastatic breast cancer (MBC). Indeed, pivotal phase III randomized clinical trials [3,4,5,6,7,8,9,10,11] demonstrated the efficacy of the combination of CDK4/6i and AIs or fulvestrant in prolonging progression free survival (PFS) in the first- and second-line settings, which often translated into a benefit in overall survival (OS) as well [12,13,14]. In the early setting, a recent interim analysis of PALLAS, a multicenter phase III trial assessing the efficacy of adding two years of palbociclib to adjuvant ET, showed the lack of a significant improvement in invasive disease-free survival (IDFS), the primary endpoint of the study [15]. On the other hand, the multicenter phase III monarchE study demonstrated the superiority of the addition of adjuvant abemaciclib to ET in improving IDFS in patients with high-risk early disease [16].

Despite the significant improvements in survival determined by CDK4/6i in patients with HR+/HER2− MBC, resistance represents a major clinical challenge. Resistance might present immediately after treatment initiation (de novo or primary resistance) or after evidence of initial clinical benefit (acquired or secondary resistance). Although a consensus on the definition of primary resistance to CDK4/6i is lacking, this may be defined as disease progression within 3–6 months of treatment initiation. Primary or de novo resistance occurs in about 15% of patients receiving CDK4/6i with AIs, and about 30% of those receiving CDK4/6i with fulvestrant. Additionally, acquired resistance develops in nearly all patients with MBC [17,18]. One of the main goals of translational research in the CDK4/6i space is the identification of biomarkers of primary/de novo or acquired resistance to personalize therapeutic strategies.

In a population with MBC, collecting metastatic tissue before, during or after treatment is challenging due to the difficulties, potential side effects and patient uneasiness in obtaining a biopsy of the metastatic lesions. This might represent a shortcoming for biomarker discovery. In this population several circulating biomarkers, including circulating tumor DNA (ctDNA) and RNA (ctRNA), microRNA (miRNA), exosomes, proteins but also metabolites and circulating tumor cells (CTCs), are currently being investigated for their potential to identify patients with primary resistance, monitor the effects of treatments and also direct later therapies with a simple withdrawal of peripheral blood (Figure 1).

A detailed and comprehensive review on biomarkers of CDK4/6i resistance has been recently published by our group [19]. In the present review, we focus on circulating biomarkers and highlight all studies and recent advancements on their potential clinical value in patients with CDK4/6i-treated HR+/HER2− MBC. Some of the studies described are yet to be published in full, but derive from pivotal clinical trials on CDK4/6i in HR+/HER2− BC and have been presented in the form of abstracts at major international meetings. Table 1 illustrates the main clinical trials investigating CDK4/6i in HR+/HER2− BC, for which correlative studies on circulating biomarkers are available.

Table 2 illustrates the correlative studies from clinical trials on CDK4/6i in HR+/HER2− MBC and the main circulating biomarkers.

To date, studies failed to identify clear and strong predictive circulating biomarkers that might help discriminating patients with different benefits from CDK4/6i. However, some were able to identify markers that might predict poor outcome (i.e., prognostic markers) in patients treated with CDK4/6i.

## 2. Biomarkers of Resistance to CDK4/6i on ctDNA

CtDNA is part of the cell-free DNA and can be identified in the blood of the majority of patients with MBC [35]. It represents a source of tumor DNA that can be assessed for a variety of tumor genetic and epigenetic alterations [36]. Mutations, copy number variations and DNA-methylation alterations, but also the fraction of DNA deriving from the tumor (tumor fraction) as such might inform on tumor biology, disease extent and patients outcome [37].

In CDK4/6i-treated patients, ctDNA has been exploited to detect biomarkers of de novo resistance, to assess dynamics of genomic alterations during CDK4/6i administration and to evaluate biomarkers of acquired resistance at the time of tumor progression.

### 2.1. Biomarkers of De Novo CDK4/6i Resistance on ctDNA

CDK4/6i mechanism of action is centered on Retinoblastoma protein (Rb), the product of the retinoblastoma tumor susceptibility gene (*RB1*), which is the main target of the CDK4/6-cyclin D complex and has a critical role in cell cycle regulation [38]. Preclinical evidence suggests that alterations in *RB1* or other cell cycle regulators such as amplification of cyclin E gene, *CCNE1*, may confer resistance to CDK4/6i [17,39,40,41]. Hence, alterations of genes involved in cell cycle regulation have been tested on ctDNA for their associations with outcomes in patients treated with CDK4/6i. Loss of *RB1* (17.3% of total patients) was associated with poorer PFS in patients treated with palbociclib and fulvestrant in PALOMA-3 trial [23]. Similarly, loss and/or loss of heterozygosity of cyclin-dependent kinase inhibitor 2A (*CDKN2A*) (22% of total patients) and copy number gains in *CDK4* and *CCNE1* (both present in <5% of patients) were associated with worse prognosis. These associations were significant at univariate, but not multivariate analysis, and no interaction with treatment was found. However, associations between loss of *RB1* and gains in *CCNE1* or *CDK4* and PFS were observed in the palbociclib and fulvestrant, but not in the placebo arm [23]. In a combined analysis of three different phase III randomized trials of ribociclib plus ET, namely the MONALEESA-2, -3 and -7, ctDNA obtained before treatment from 1503 patients was analyzed by next generation sequencing with a targeted panel containing 557-genes [32]. Patients with *RB1* wild-type ctDNA tended to have more PFS benefit from ribociclib compared to patients with *RB1*-mutant ctDNA (1.7% of total) [32]. In addition, patients with alterations in *CDKN2A/2B/2C* (2.3% of total tumors) derived limited benefit from the addition of ribociclib to ET [32]. A trend towards limited benefit from ribociclib was observed in patients with alteration in cell-cycle related genes in the MONALEESA-3 trial [34]. BioItaLEE (NCT03439046) is a single-arm, phase IIIb trial of patients with MBC receiving ribociclib and letrozole as first-line. The trial analyzed ctDNA alterations before starting treatment and assessed their associations with clinical outcome [42]. Alterations within the CDK4/6-Rb pathway genes, including *CDK4* and *6, CCND1*, *CDKN2A* and *RB1*, were associated with early progression [42]. All these data highlight the potential for alterations in genes within the CDK4/6-Rb pathway to serve as circulating biomarkers of palbociclib resistance. However, more data are needed to establish their clinical utility.

Phosphatidylinositol-4,5-bisphosphate 3-kinase, catalytic subunit alpha (*PIK3CA*) is the most frequently altered gene in HR+/HER2− MBC [43]. Preclinical studies suggest the phosphatidylinositol 3-kinase (PI3K)/mammalian target of rapamycin (mTOR)/protein kinase B (PKB, AKT) pathway might have a role in CDK4/6i resistance [39,44,45,46,47]. Thus, several studies analyzed ctDNA alterations within the PI3K pathway and their association with response to CDK4/6i. Mutations in *PIK3CA* were found in ctDNA of 33%, 35%, 33% and 40.3% of patients in the PALOMA-3 [24], MONALEESA-3 [34], MONALEESA-2 [33] and MONARCH-2 [31], respectively. CDK4/6i similarly prolonged PFS in patients with *PIK3CA* mutated or wt ctDNA, suggesting the lack of a role for baseline *PIK3CA* mutations in predicting benefit from CDK4/6i. On the other hand, the prognostic role in patients treated with CDK4/6i remains unclear. Indeed, patients with *PIK3CA*-mutant and -wild-type (wt) tumors treated with palbociclib and fulvestrant within PALOMA-3 showed similar PFS (median PFS of 9.5 and 9.9 months, respectively) [24], while among patients receiving ribociclib and fulvestrant within MONALEESA-3, those with *PIK3CA*-mutant ctDNA showed numerically shorter PFS compared to *PIK3CA*-wt (16.4 vs. 22.3 months, respectively) [34]. Similarly, in MONALEESA-2 patients receiving ribociclib and letrozole with *PIK3CA*-wt, ctDNA had a median PFS of 29.6 months compared to 19.2 in those with *PIK3CA*-mutant [33]. The tumor suppressor phosphatase and tensin homolog (*PTEN)* is a negative regulator of the PI3K signaling. Loss of *PTEN* and *AKT1* amplification or *AKT1* activating mutations in tumor samples have been associated with resistance to CDK4/6i [48,49]. Consistent with these data, loss of *PTEN* in ctDNA was associated with worse PFS in patients treated with palbociclib and fulvestrant within PALOMA-3 [23]. On the other hand, in the MONALEESA-2 -3 and -7 pooled analysis patients with *AKT1* alterations, particularly *AKT1 E17K,* showed increased benefit from ribociclib over placebo [32]. Alterations of *PTEN* and *AKT1* might also be relevant for therapeutic strategies subsequent to CDK4/6i, since *PTEN* loss might mediate resistance to PI3Kα inhibitors [50,51] and the AKT inhibitor capivasertib demonstrated clinical activity in *AKT1 E17K*-mutant MBC [52].

Additional alterations that have been tested on ctDNA for their association with outcome include those in the estrogen receptor *(ESR1)*, fibroblast growth factor receptor 1 (*FGFR1)* and tumor protein 53 (*TP53)* genes. Mutations in *ESR1* are frequently acquired in HR+/HER2− MBC following ET, particularly AIs [53]. *ESR1* mutations were detected in 25.3%, 14% and 64.4% of plasma samples from patients enrolled in PALOMA-3 [25] MONALEESA-3 [34] and MONARCH-2 [31] trials, respectively. In PALOMA-3, they did not predict a benefit from palbociclib [25]. In MONALEESA-3, there was a trend toward increased PFS benefit from ribociclib vs. placebo for patients with *ESR1*-mutant tumors [34], and in MONARCH-2, there was a numerically greater benefit from abemaciclib plus fulvestrant in patients with *ESR1*-mutant tumors [31]. Results from the PEARL, a phase III multicenter study, have recently been published [22]. In this study, patients with HR+/HER2− MBC resistant to AIs were randomized to receive palbociclib and exemestane vs. capecitabine (cohort 1). After the discovery that *ESR1* mutations might be responsible for resistance to AIs, it was amended to include cohort 2, randomizing patients to palbociclib and fulvestrant vs. capecitabine. The trial demonstrated the non-superiority of palbociclib and ET vs. capecitabine in both cohorts [22]. The prognostic role of *ESR1* in patients receiving CDK4/6i is yet to be established as well. Median PFS for patients receiving fulvestrant and palbociclib in PALOMA-3 did not differ according to *ESR1* status (9.4 months and 9.5 for ctDNA *ESR1*-mutant and wt, respectively) [25]. In MONARCH-2, patients with *ESR1*-mutant ctDNA receiving abemaciclib and fulvestrant demonstrated a median PFS of 21.9 months vs. 16.3 of those with *ESR1*-wt [31], while in MONALEESA-3, patients receiving ribociclib and fulvestrant had median PFS of 9.3 and 22.3 months for *ESR1*-mutant and -wt ctDNA, respectively [34]. However, MONALEESA-3 analysis was presented irrespective of the line of treatment, and *ESR1* mutations were more frequently observed in patients receiving treatment with ribociclib as second line compared to first line (24.6% vs. 4.3%, respectively) [34]. PADA-1 trial (NCT03079011) is a phase III trial evaluating the utility of monitoring ctDNA for the onset of *ESR1* mutations in patients receiving palbociclib plus AIs in first line [54]. The prognostic impact of *ESR1* was recently presented. Of 1017 patients analyzed, 33 had detectable circulating mutations in *ESR1* at inclusion (3.2%) and showed a significantly shorter PFS [54].

Altered FGFR-1 signaling was shown to mediate CDK4/6i resistance [55]. In PALOMA-3, baseline *FGFR1* gain on ctDNA was found to be associated with worse PFS in both the palbociclib and fulvestrant and the placebo and fulvestrant arms [23]. Associations remained significant at a multivariate analysis, but no interaction with treatment was found [23]. In MONALEESA-2, patients treated with ribociclib plus letrozole with *FGFR1* amplification on ctDNA (5% of total patients) experienced shorter PFS compared to those with *FGFR1*-wt ctDNA [55]. However, in MONALEESA-3 and MONALEESA-2, the benefit from ribociclib was observed regardless of ctDNA *FGFR1* alterations found at baseline [33,34]. These data point toward a prognostic rather than predictive role of *FGFR1.* Similarly, *TP53* alterations seem to identify patients at risk of early progression regardless of CDK4/6i treatment. Indeed, in both PALOMA-3 and BioItaLEE, baseline ctDNA *TP53* alterations were associated with significantly shorter PFS [23,42], but in PALOMA-3 associations with PFS were found in both palbociclib and placebo arms, and no interaction with treatment was found [23]. Additionally, in MONALEESA-3 and MONALEESA-2, ribociclib benefit was observed independently of *TP53* alterations [33,34].

Tumors often harbor multiple gene alterations; therefore, evaluating the correlations between a single genomic alterations and response to CDK4/6i might be challenging. Additionally, estimation of copy number variations, particularly loss, on ctDNA is technically challenging. This, coupled with the low prevalence of some alterations such as *RB1* in HR+/HER2− BC, might make associations with patients outcome and interaction with treatment difficult to establish. In PALOMA-3, authors have focused on circulating tumor fraction assessment to identify patients with poor outcome [23]. They found that tumor fraction was associated with adverse PFS in both palbociclib and placebo arms and was independently associated with outcome at the multivariate analysis. Intriguingly, at the multivariate analysis, among all alterations found to be associated with PFS, only *TP53* alterations and *FGFR1* gain maintained an independent prognostic role [23], suggesting that tumor fraction might be a confounding factor during the evaluation of circulating biomarkers on ctDNA.

### 2.2. Dynamics of ctDNA Biomarkers

Other than assessing baseline biomarkers, an alternative strategy for predicting patients outcome might be to monitor early changes in selected markers. Circulating biomarkers are particularly helpful to this end for their ease to be repeated over time.

In PALOMA−3, *PIK3CA* and *ESR1* mutations were analyzed both at baseline and after 15 days of palbociclib plus fulvestrant treatment [26]. After 15 days, relative changes from baseline in *PIK3CA* mutation levels were strongly predictive of PFS, while changes in *ESR1* mutations were of limited prediction [26]. However, only 22% and 25.6% of patients analyzed in this study had *PIK3CA* and *ESR1* ctDNA mutations, respectively [26]. To bypass this problem, in the ALCINA study (NCT02866149), 25 HR+/HER2− MBC patients receiving palbociclib and fulvestrant were assessed for ctDNA by droplet-digital PCR (ddPCR) based on somatic mutations found on their primary or metastatic tissue that could be tracked in circulating DNA, the majority being found in *PIK3CA* (*n* = 21), but also in *TP53* (*n* = 2) and *AKT1* (*n* = 2) genes [56]. At baseline, 84% of patients had detectable ctDNA levels, but these were not prognostic. Clearance of ctDNA observed at day 30 was associated with longer PFS, while an increase of ctDNA levels at day 30 compared to baseline was predictive of shorter PFS. Of note, in this study, dynamics at day 15 had no prognostic impact [56]. To overcome the need of prioritizing mutations based on tumor tissue genomics, Martínez-Sáez and colleagues sequenced plasma samples from 45 patients with CDK4/6i-treated HR+/HER2− MBC using the standardized Guardant360 assay [57]. Of these, 43 (96%) had ctDNA detectable at some level. Authors found that the mean variant allele fraction ratio (mVAFR), calculated between the first day of cycle 2 and baseline, was significantly associated with PFS. However, they found no association with baseline and on-treatment VAF or absolute changes in VAF [57]. The different populations and methodologies and the lack of a univocal definition of tumor fraction might explain at least in part the different results across the studies.

### 2.3. Circulating Biomarkers of Acquired Resistance on ctDNA

In MBC, acquired resistance to CDK4/6i and ET near-inevitably occurs. At the time of progression tumors might acquire new genetic alterations that can be analyzed on ctDNA samples. Analysis of these alterations might potentially help in deciphering the resistance mechanisms and also directing subsequent therapies. In PALOMA-3, analyzing paired ctDNA from baseline and end-of-treatment samples from 195 patients by target sequencing, newly acquired mutations in *RB1*, *PIK3CA* and *ESR1* were identified at the time of progression [27]. *RB1* acquired mutations were observed only in patients receiving palbociclib plus fulvestrant (4.7% of tumors), while mutations in *PIK3CA* and *ESR1* were acquired in both treatment groups (8.2% and 12.8% of patients, respectively), particularly *PIK3CA E542K* and *ESR1 Y537S* and *D538G* mutations [27]. Existing therapeutic strategies after failure of CDK4/6i include chemotherapy, ET alone or ET combined with targeted agents. Targeting the PI3K/mTOR/AKT pathway, continuing CDK4/6i after progression, or new more powerful SERDs are among the therapeutic strategies currently being evaluated after CDK4/6i progression [19]. The potential clinical utility of testing targetable genomic alterations in ctDNA of patients with MBC was recently reported [58], but whether *PIK3CA*, *ESR1*, *RB1* or alterations in other genes at the time of progression to CDK4/6i might help personalize subsequent therapeutic strategies remain to be established in future studies.

## 3. The Role of Thymidine Kinase-1 Serum Activity

Thymidine kinase 1 (TK1) is an enzyme involved in the DNA salvage pathway which catalyzes the phosphorylation of thymidine to the monophosphate form (dTMP), which is then further phosphorylated to deoxy thymidine triphosphate (dTTP) before being incorporated into DNA [59]. TK1 has also a crucial role in DNA damage repair, being essential in replacing the pool of dTTP in case of cellular DNA damage [59]. TK1 is expressed mainly in dividing cells, increasing during the G1-S phase and being degraded after cell division [59]. Therefore, it could be deemed as a marker of cell proliferation. Moreover, the synthesis of TK1 is regulated by the E2F transcription factors whose activation is regulated by CDK4/6−Rb pathway, suggesting a potential role for TK1 in monitoring the activity and efficacy of CDK4/6i [60].

Several studies showed the elevated levels of TK1 in many cancer types, including breast [61]. TK1 can be assessed in plasma or serum samples from patients with BC [62,63] and levels of circulating TK1 activity (TKa), both at baseline and during treatments, were prognostic in ET−treated MBC patients [64,65].

The first report on serum TKa in patients treated with CDK4/6i derives from NeoPalAna, a pre-operative trial in which patients with stage II and III HR+/HER2− BC received anastrozole alone, followed by palbociclib and anastrozole for four cycles, followed by the pre-surgical washout of palbociclib with the exception of eight patients in which the combination was received until surgery [20]. During treatment with anastrozole alone, there was no significant change in TKa, while a marked reduction was observed after two weeks from palbociclib initiation. During preoperative washout, TKa increased significantly, but remained low in those who continued palbociclib until surgery [28]. There was a high correlation between changes in serum TKa and tumor Ki−67. These data suggested a pharmacodynamic role for serum TKa in palbociclib-treated patients [28].

The hypothesis that baseline TKa or early changes in TKa levels might be prognostic in patients with HR+/HER2− MBC treated with CDK4/6i was then explored in TREnd [29]. In this trial, comparing the efficacy and safety of single-agent palbociclib vs. palbociclib plus the ET previously received for MBC, baseline TKa was not a poor prognostic factor. However, patients showing an increase in TKa after one month of treatment had a worse PFS compared to those without increase, suggesting TKa dynamics might have a role as an early marker of resistance to palbociclib [29].

The prognostic role of TKa was further evaluated in 103 plasma samples obtained from ER+/HER2- MBC patients treated with ET and palbociclib within ALCINA study [66]. In this study, baseline TKa was an independent poor prognostic marker of PFS and OS. Additionally, TKa at 4-week was associated with OS [66], while adding the changes in TKa at 4 weeks compared to baseline did not further increase prediction.

Lately, results from PYTHIA (NTC02536742), a biomarker discovery phase II single-arm study of fulvestrant and palbociclib in 122 women with HR+/HER2− MBC progressing on prior ET, were presented [67]. TKa was assessed at baseline, after 15 days of treatment and before initiating cycle 2. At each timepoint higher TKa was significantly and independently associated with PFS [67]. Interestingly, the group of patients who, at day 15, did not experience a drop in TKa below the limit of detection of the assay, suggesting an incomplete biomarker response, was enriched for patients with de novo resistance to palbociclib plus fulvestrant [67].

There is still uncertainty regarding the optimal method for quantifying TK1 levels and activity, the reproducibility or the optimal cut-off to be used. However, serum TKa can be easily measured through a peripheral blood draw and quantified via ELISA-based assays, thus representing a non-invasive and cost-effective way of estimating prognosis at baseline, as well as monitoring treatment response in patients with CDK4/6i-treated HR+/HER2− MBC [68]. Additional studies are needed to establish its clinical utility.

## 4. Additional Circulating Biomarkers

CTCs are cancer cells found in the bloodstream after detaching from primary and/or metastatic tumors. Their concentration is usually very low making their detection quite challenging; despite this, the prognostic role of CTCs in patients with MBC was extensively demonstrated [69,70]. Additionally, molecular characterization of CTCs might increase their clinical validity [71]. The prognostic value of CTC was recently analyzed in patients enrolled in a translational sub-study of TREnd [30]. CTCs count after the first cycle of palbociclib (T1), but not at baseline was prognostic in terms of PFS. Additionally, the dynamics of CTCs count at T1 was prognostic. Indeed, patients with an increase of three or more CTCs at T1 experienced shorter PFS compared to those without increase. Intriguingly, patients whose CTCs showed detectable expression of *RB1* at any time-point had better, although not statistically significant, outcomes compared to those with undetectable levels [30].

Exosomes are micro-vesicles ranging 40–150 nm in size carrying proteins, RNA and DNA. They are actively released from cancer cells and can be found in patients’ blood [72]. *TK1*, *CDK4*, *CDK6* and *CDK9* expression have been analyzed by ddPCR on RNA extracted from exosomes in 40 patients with HR+/HER2− MBC before the administration of palbociclib and ET (T0), and after 3 months of treatment (T1) [73]. *CDK4* levels at T0 correlated with longer PFS, while a significant increase of *TK1* and *CDK9* at T1 compared to T0 was found in patients with progressive disease [73].

DNA methylation is an epigenetic phenomenon in which a methyl group is added to the fifth carbon of the cytosine residue by DNA methyltransferases (DNMTs), predominantly in a CpG dinucleotide context, and is associated with gene silencing [74]. A recent study tested the feasibility of characterizing the epigenetic status of *ESR1* by assessing the methylation status of its two main promoters, namely promA and promB, using methylation-specific ddPCR [75]. CtDNA from 49 women with HR+/HER2− MBC predominantly treated with CDK4/6i and ET was analyzed before starting treatment and after 3 months, at the time of restaging. While baseline methylation levels of both promA and promB were not associated with PFS, an increase in promB or in either promA or promB at restaging was associated with a significantly worse prognosis [75].

MiRNA are small non-coding RNAs that, by modulating specific target mRNA, might play a major role in physiological or pathological processes, including CDK4/6i resistance [76,77]. Indeed, miR-223 was shown to be a modulator of CDK4/6i response in vitro and in vivo [76], and the exosomal miR-432-5p has been implicated in resistance to CDK4/6i [77]. However, data on circulating miRNA in patients with HR+/HER2− BC treated with CDK4/6i is lacking.

## 5. Conclusions

Biomarkers are key to personalize medicine. Notwithstanding the significant and rigorous efforts that have been made to identify potential biomarkers of de novo or acquired resistance to CDK4/6i, none of the investigated markers has been implemented in the clinical practice yet. Research efforts are focusing on circulating biomarkers given their numerous advantages for both clinical development and application: they can be obtained non-invasively through a simple withdrawal of peripheral blood in virtually all patients with MBC; they are easy to be repeated over time allowing real-time monitoring of therapy; they might be representative of all metastatic sites and genetic clones within a tumor. However, one of the main drawbacks in the development of circulating biomarkers includes their low concentrations, which challenges the creation of both sensitive and precise tests. Studies showed discrepant results probably due to the different technologies and methodologies used, the different study populations analyzed and possibly the type of CDK4/6i administered. Therefore, further studies are needed to identify clinically useful biomarkers of CDK4/6i response. Future efforts to address these issues must be made before circulating biomarkers might enter into the clinical management of patients with HR+/HER2− MBC.

## Figures and Tables

**Figure 1 cancers-13-02640-f001:**
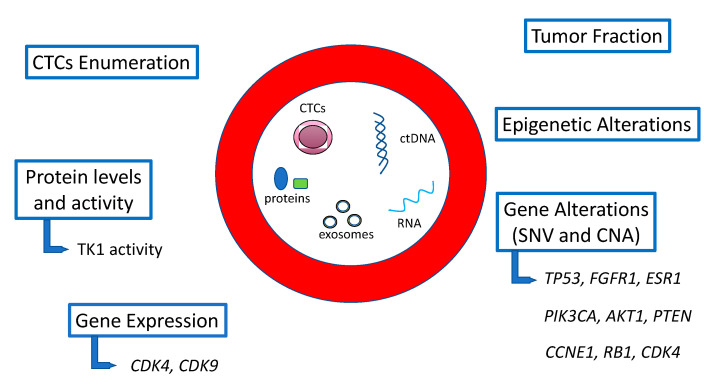
Schematic illustration of some of the potential circulating biomarkers of response to CDK4/6i in HR+/HER2− BC discussed in this review (TK1 = thymidine kinase 1; SNV = single nucleotide variations; CNA = copy number alterations).

**Table 1 cancers-13-02640-t001:** Clinical trials on CDK4/6i in HR+/HER2− BC with correlative studies on circulating biomarkers.

Name	Phase	Setting	Treatments	Results
PALOMA-3 [4]	3	MBC II line	Pal and Ful vs. Plb and Ful	9.2 vs. 3.8 months PFS HR 0.42, 95% CI 0.32–0.56
NEOPALANA [20]	2	Neoadjuvant	Ana → Ana and Pal → Ana	87% vs. 26% CCCA
TREND [21]	2	MBC ≥ II line	Pal vs. Pal and ET earlier line	60% vs. 54% CBR
PEARL [22]	3	MBC (AI-resistant)	Cohort 1: Pal and Exe vs. Cap Cohort 2: Pal and Ful vs. Cap	Cohort 2: 7.5 vs. 10.0 months PFS HR 1.13, 95% CI 0.85–1.50 Cohorts 1 + 2, *ESR1* wt: 8.0 vs. 10.6 months PFS HR: 1.11, 95% CI: 0.87–1.41
MONARCH-2 [9]	3	MBC II line	Abema and Ful vs. Plb + Ful	16.4 vs. 9.3 months PFS HR 0.553, 95% CI 0.449–0.681
MONALEESA-2 [6]	3	MBC I line	Ribo and Let vs. Plb and Let	NR vs. 14.7 months PFS HR 0.56, 95% CI 0.43–0.72
MONALEESA-3 [10]	3	MBC I-II line	Ribo and Ful vs. Plb and Ful	20.5 vs. 12.8 months PFS HR 0.593, 95% CI 0.480–0.732
MONALEESA-7 [11]	3	MBC I line	Ribo and Tam or NSAI and Gos vs. Plb and Tam or NSAI and Gos	23.8 vs. 13.0 months PFS HR 0.55, 95% CI 0.44–0.69

PFS = progression free survival; CCCA = Ki67 ≤ 2.7% or complete cell cycle arrest; CBR = clinical benefit rate; HR = hazard ratio; ET = endocrine therapy; Pal = Palbociclib; Ribo = Ribociclib; Abema = Abemaciclib; Ful = Fulvestrant; Ana = Anastrazole; Exe = Exemestane; Let = letrozole; Tam = Tamoxifen; Gos = Goserelin; Plb = Placebo; Cap = Capecitabine; NSAI = Non-steroidal aromatase inhibitor.

**Table 2 cancers-13-02640-t002:** Correlative studies on circulating biomarkers for main clinical trials on CDK4/6i.

Clinical trial	Correlative Studies	Methods	Main Circulating Biomarkers
PALOMA-3 [4]	O’Leary et al. [23]	Targeted panel	*TP53*, *FGFR1*, TF
	Cristofanilli et al. [24]	ddPCR	*PIK3CA*
	Fribbens et al. [25]	ddPCR	*ESR1*
	O’Leary et al. [26]	ddPCR	*PIK3CA, ESR1* dynamics
	O’Leary et al. [27]	Targeted panel	acquired mutations
NEOPALANA [20]	Bagegni et al. [28]	Divitum	TKa
TREND [21]	McCartney et al. [29]	Divitum	TKa
	Galardi et al. [30]	CellSearch	CTC
MONARCH-2 [9]	Tolaney et al. [31]	ddPCR	*PIK3CA*, *ESR1*
MONALEESA-2 [6]	Andre et al. [32]	Targeted panel	*CHD4*, *ATM*, *FRS2*, *PRKCA*, *CDKN2A/2B/2C*, *AKT1*
	Hortobagyi et al. [33]	Targeted panel	*PIK3CA*, *TP53*, RTK
MONALEESA-3 [10]	Andre et al. [32]	Targeted panel	*CHD4*, *ATM*, *FRS2*, *PRKCA*, *CDKN2A/2B/2C*, *AKT1*
	Neven et al. [34]	Targeted panel	*PIK3CA, ESR1, TP53, FGFR1*, CC, RTK
MONALEESA-7 [11]	Andre et al. [32]	Targeted panel	*CHD4*, *ATM*, *FRS2*, *PRKCA*, *CDKN2A/2B/2C*, *AKT1*

TF = tumor fraction; TKa = Thymidine kinase activity; ddPCR = droplet digital PCR; CTC = circulating tumor cells; CC = cell cycle associated genes; RTK = genes involved in receptor tyrosine kinase.

## Data Availability

Not applicable.

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
