# Peer review of "Circulating Biomarkers of CDK4/6 Inhibitors Response in Hormone Receptor Positive and HER2 Negative Breast Cancer"

_cancers, 2021, doi:10.3390/cancers13112640_

Round 1
Reviewer 1 Report
The paper will be interesting to readers for circulating markers may offer a real-time investigation tool beyond genomics and clinical characteristics for HR(+) advanced breast cancer patients, and future clinical risk stratification may be best determined with a composite score incorporating clinical features together with tumor- and blood-based markers.
It will be appreciated if the authors offer a Table that summarizes the clinical outcomes data with validation.
Reviewer 2 Report
With this work the authors describe the latest advancements in the development of circulating biomarkers for patients with CDK4/6i-72 treated HR+/HER2- metastatic breast cancer.
The review by itself is well written and comprehensive, well detailed in the analysis of the existing clinical trials.
However, as correctly stated by the authors, "A detailed and comprehensive review on biomarkers of CDK4/6i resistance has been recently published by our group". I find the present manuscript a repetition of the previously published one, without introducing novelty or additional information that worth a second publication.
If the difference between the present work and the previously published one is the focus on metastatic breast cancer, thus I suggest to mark this difference in a clearer way, highlighting the differences of CDK4/6i resistance biomarkers in MBC compared to primary breast cancer and to solely focus on metastatic breast cancer biomarkers.
The focus on metastatic breast cancer shuold then be mentioned in the title too.
Finally, considering the vast number of possible biomarkers, the authors should also mention microRNAs as they are rising as potential biomarkers for several applications.
Reviewer 3 Report
In this review, the authors organized the circulation biomarkers of CDK4/6 inhibitors in hormone receptor positive and her2 negative breast cancer patients. The authors focused on the results of circulating ctDNA. However, the low concentration of ctDNA may loss the sensitivity. Additionally, the mutation in some biomarkers, such as RB, may not entirely caused by CDK4/6 inhibitors resistance. The strategy for addressing the points should be add in the last section.
Round 2
Reviewer 2 Report
The Authors have satisfied all the requiremens and addressed all the reviewer suggestions. Thus, I recommend acceptance of the present version of the manuscript.